# Exploring Hypercoagulability in Post-COVID Syndrome (PCS): An Attempt at Unraveling the Endothelial Dysfunction

**DOI:** 10.3390/jcm14030789

**Published:** 2025-01-25

**Authors:** Maxim Muys, Anne Demulder, Tatiana Besse-Hammer, Nathalie Ghorra, Laurence Rozen

**Affiliations:** 1Laboratory of Hematology, CHU Brugmann, Université Libre de Bruxelles (ULB), 1020 Brussels, Belgium; anne.demulder@chu-brugmann.be (A.D.); laurence.rozen@chu-brugmann.be (L.R.); 2Clinical Research Unit, CHU Brugmann, Université Libre de Bruxelles (ULB), 1020 Brussels, Belgium; tatiana.besse-hammer@chu-brugmann.be; 3Laboratory of Immunology, CHU Brugmann, Université Libre de Bruxelles (ULB), 1020 Brussels, Belgium; nathalie.ghorra@chu-brugmann.be

**Keywords:** long COVID, post-COVID syndrome, post-acute sequelae of SARS-CoV-2 infection, hypercoagulable state, thrombin generation, thromboembolic risk, thrombomodulin, ADAMTS-13, von Willebrand factor

## Abstract

**Background**: The lingering effects of SARS-CoV-2 infection, collectively known as post-COVID syndrome (PCS), affect a significant proportion of recovered patients, manifesting as persistent symptoms like fatigue, cognitive dysfunction, and exercise intolerance. Increasing evidence suggests that endothelial dysfunction and coagulation abnormalities play a central role in PCS pathophysiology. This study investigates hypercoagulability and endothelial dysfunction in PCS through thrombin generation and the von Willebrand factor (VWF)/ADAMTS13 axis. **Methods**: Plasma samples from 97 PCS patients recruited since October 2020 by the clinical research unit of the Brugmann University Hospital were analyzed. A thrombin generation test was performed on a St-Genesia^®^ analyzer (Stago) using the Thromboscreen kit; VWF antigen was determined on a CS-2500 analyzer (Siemens); and ADAMTS-13 activity was determined using an ELISA kit (Technozym^®^) on an ElX808 plate reader. **Results**: Thrombin generation testing revealed elevated thrombin production in PCS patients, particularly when thrombomodulin was included. Although most PCS patients showed normalized VWF/ADAMTS13 ratios, 11.3% exhibited elevated ratios (≥1.5), associated with advanced age. **Conclusions**: Patients with PCS show a consistent pattern of prolonged thrombo-inflammatory dysregulation, highlighted by elevated in vitro thrombin generation and the persistence of abnormal VWF/ADAMTS-13 ratios in a subset of patients.

## 1. Introduction

The emergence of the COVID-19 pandemic has led to significant global health consequences, not only from the acute effects of the SARS-CoV-2 virus but also from its long-term impact. A substantial proportion of individuals who recover from the acute phase of infection experience lingering health issues, collectively known as long COVID, post-acute sequelae of SARS-CoV-2 infection (PASC), or post-COVID syndrome (PCS). It is estimated that approximately 30% of patients show the presence of post-COVID symptoms two years after COVID-19 infection [1]. These symptoms, which include fatigue, shortness of breath, cognitive dysfunction, and decreased exercise tolerance, can severely affect an individual’s quality of life and are often heterogeneous in presentation [2,3].

PCS has garnered significant interest within the scientific community due to its complex and multifaceted nature. Growing evidence suggests that endothelial dysfunction and persistent coagulation abnormalities may be central to the condition. Numerous studies have already demonstrated that acute SARS-CoV-2 infection triggers a hypercoagulable state with increased thrombin generation, especially noticeable in hospitalized patients [4]. Elevated D-dimer levels, which correlate with poor outcomes, along with increased levels of factor VIII and von Willebrand factor (VWF), further support this hypercoagulability hypothesis [5,6]. The hypercoagulable state seen in acute infection, highlighted by excess thrombin generation, can extend into the recovery phase, contributing to prolonged symptoms in patients with PCS [7]. Recent studies suggest that the persistence of SARS-CoV-2 viral particles in tissues, including endothelial cells, may continue to drive chronic inflammation and endothelial damage in patients with PCS. The viral spike protein induces clumping and clotting of red blood cells and platelets, further exacerbating endothelial injury and promoting the formation of microthrombi [8,9].

Of particular interest is the role of thrombomodulin (TM), a membrane-bound protein expressed on endothelial cells that is essential for the proper functioning of the protein C anticoagulant pathway. TM binds to thrombin, converting it from a procoagulant enzyme into an activator of protein C. This activation of protein C subsequently suppresses clotting by inactivating factor Va and factor VIIIa, thereby limiting further thrombin generation and promoting fibrinolysis. Impaired TM function, as suggested by the reduced inhibition of thrombin generation, may contribute to the hypercoagulable state observed in PCS patients [10].

In addition to TM, other markers of endothelial health, such as von Willebrand factor activity and ADAMTS-13 activity, are important in assessing endothelial dysfunction. VWF, a large multimeric glycoprotein, is crucial for platelet adhesion and aggregation at sites of vascular injury. ADAMTS-13, a protease that cleaves VWF multimers, plays a key role in regulating VWF activity. An imbalance in the VWF/ADAMTS-13 axis has been implicated in the pathogenesis of thrombotic disorders, including thrombotic thrombocytopenic purpura (TTP), and may also be relevant in the context of PCS [11].

This study aims to evaluate coagulation disorders in patients diagnosed with PCS through a detailed analysis of thrombin generation and endothelial health markers. By using a thrombin generation test with TM, alongside assessments of VWF antigen and ADAMTS-13 activity, we seek to investigate the extent of coagulation abnormalities and endothelial dysfunction in PCS patients.

## 2. Materials and Methods

Between October 2020 and July 2023, patients reporting lasting effects of their SARS-CoV-2 infection could present themselves for a consultation at the Clinical Research Unit of Brugmann University Hospital and be enrolled in a study focusing on the prolonged impact of COVID-19 on the central nervous system. Each patient underwent a thorough neurological and psychiatric evaluation to document their reported symptoms, and information on their medical history, comorbidities, and current treatments was gathered. Our study focused on the hemostatic imbalance of PCS patients enrolled in this study. During the initial consultation, a whole blood sample was collected in a 3.2% sodium-citrate tube. Follow-up visits were scheduled a year later, with frequency adjusted based on symptom persistence. Informed written consent was obtained from all participants and ethical approval was granted by the Brugmann University Hospital Ethics Committee (BUN: B0772022000066). Patients were included based on the WHO definition of long COVID (post COVID-19 condition) and needed to meet the following criteria:Symptom onset within three months following a suspected or confirmed SARS-CoV-2 infection.Symptoms lasting for at least two months.No other condition that could account for the symptoms.

PCS patients were compared to historical cohorts, including healthy adults used as a control group [12] and individuals with acute COVID [4].

Samples were frozen at −80 °C after double centrifugation at 1900× *g* for 15 min. Before analysis, samples were thawed in a water bath at 37 °C for 5 min.

### 2.1. Laboratory Tests

Thrombin generation test (TGT) analysis was performed on a St-Genesia analyzer on PCS patient plasmas, both with and without TM using the Thromboscreen reagent that includes an intermediate tissue factor level (Diagnostica Stago, Asnières, France). Analysis of TGT data covered lag time, peak height, time to peak, endogenous thrombin potential (ETP), ETP inhibition, and velocity index. We mostly focused on ETP as it is the most widely studied parameter in TGT and has significant clinical value, offering a comprehensive assessment of an individual’s thrombotic potential. ETP represents the total amount of thrombin generated during TGT by calculating the area under the thrombin generation curve (AUC). A higher ETP indicates increased thrombin production, suggesting a higher thrombotic risk. ETP is useful for evaluating the risk of thrombotic events, guiding treatment decisions, and monitoring coagulation disorders in various conditions, including PCS. ETP inhibition was determined by comparing test results with and without TM, offering insight into anticoagulation from the protein C/protein S (PC/PS) complex.

VWF antigen and ADAMTS13 activity were measured using the VWF Ag assay on the CS-2500 analyzer (Siemens, Munich, Germany) and the Technozym^®^ ADAMTS13 Activity ELISA kit, respectively. The ELISA plate was then read on the BioTek Winooski, VT, USA ElX808 plate reader.

### 2.2. Brain Imaging

As cognitive impairment is among the most common symptoms of PCS, we looked at neuroimaging techniques to better understand the pathophysiological brain changes in PCS. In collaboration with the Department of Nuclear Medicine, we examined regional cerebral perfusion alterations in individuals with PCS using scintigraphy.

### 2.3. Statistical Analysis

All statistical analyses were performed on GraphPad Prism^®^ software (version 5.01, GraphPad Software, San Diego, CA, USA). Continuous variables were presented as median values with interquartile ranges (IQRs) and compared across four groups using the Kruskal–Wallis test. In the cases where Kruskal–Wallis test were significant, Dunn’s post hoc test was performed to compare all pairs of the groups. When comparing ranks across 2 groups, we used the Mann–Whitney test. All statistical tests were two-sided, with statistical significance set at *p* ≤ 0.05.

## 3. Results

### 3.1. Demographic Data

The historical control cohorts consisted of 81 healthy adult individuals [12], used as a control group, and 23 acute COVID patients [4]. The acute COVID-19-positive patient group was further divided into two subgroups (severe and non-severe) according to clinical presentation and disease progression. The non-severe subgroup comprised 15 patients who did not require hospitalization, while the severe subgroup included 8 patients with hypoxemic pneumopathy necessitating oxygen therapy and/or CT scan evidence of lung parenchyma damage exceeding 25%. The median ages for these subgroups were 53 years (range: 46–71) and 50 years (range: 48–74), respectively. The PCS group included 97 patients, aged 22 to 80, with 76 females and 21 males. The data for each patient group, including participant ages and male-to-female ratios, are summarized in Table 1 and Figure 1.

### 3.2. Thrombin Generations Tests (TGTs)

Differences among healthy subjects, severe and non-severe acute COVID-19 patients were already described in our previous work [4]. This study focuses on assessing coagulation abnormalities in patients diagnosed with PCS (Table 2). Thrombin generation tests without TM did not reveal any statistically significant differences between the PCS cohort and the other cohorts (acute COVID and healthy subjects). Abnormalities became evident through TGT analysis in the presence of TM. When TM was included, the ETP inhibition was lower in patients with PCS compared to healthy controls, meaning the thrombin generation was notably higher in those patients, albeit to a lesser extent when compared to those with severe acute COVID-19 (Figure 2).

### 3.3. Endothelial Markers: VWF(Ag) and ADAMTS-13 Activity

To assess endothelial function, we further tested VWF antigen and ADAMTS-13 activity in 97 PCS patients. Unfortunately, these parameters were not determined in the historical cohorts, so we lack healthy control samples for VWF(Ag) and ADAMTS13 activity. Results were compared to reference values published in the literature [13]. In agreement with these values, a typical ratio would likely be around 1. A ratio above 1.5 could be a potential threshold of heigtened thrombotic risk. No statistically significant differences were observed in VWF(Ag) levels and ADAMTS13 activity, with a median of 98 IU/dL (IQR, 77–127) and 105 IU/dL (IQR, 97–112), respectively (Table 3). Nevertheless, among the 97 patients with PCS, 12 (12.4%) had an elevated level of VWF(Ag), ranging from 150 to 226 IU/dL (NR, 50–150 IU/dL). ADAMTS13 activity remained in the normal range between 74 and 131 IU/dL.

Overall, the VWF(Ag) to ADAMTS13 activity ratio did not appear elevated, with a median value of 0.91 (IQR, 0.73–1.3). However, 11% of PCS patients showed an abnormal VWF/ADAMTS13 ratio of ≥1.5, a threshold potentially indicative of heightened thrombotic risk.

We further divided our PCS cohort into two groups based on the VWF(Ag)/ADAMTS13 ratio: one group with a ratio < 1.5 and another with a ratio ≥ 1.5. Each group was analyzed separately in relation to demographic data and thrombin generation test results. The group with a ratio ≥ 1.5 was significantly older, with a median age of 54 (43–80) years compared to 47 (23–73) in the ≤1.5 group (Mann–Whitney *p*-value < 0.005). TGT parameters and BMI did not show any significant differences (Table 4). Additionally, we found that all patients with an abnormal ratio showed hypoperfusion in the frontal lobes or perfusion heterogeneity on cerebral perfusion scintigraphy.

Spearman’s rank correlation analysis also found no significant correlation between thrombin generation, VWF, ADAMTS13, and the VWF/ADAMTS13 ratio.

## 4. Discussion

Extended functional impairment with high symptom burdens well beyond the acute phase of COVID-19, often referred to as post-COVID syndrome (PCS), is estimated to affect 30% of infected individuals [1,14,15]. Central to COVID-19 pathogenesis is a thrombo-inflammatory process, with elevated levels of factor VIII, von Willebrand factor (VWF), and D-dimer [16] being key markers indicative of a hypercoagulable state. The relationship between hypercoagulability and symptoms of PCS is complex, yet it is increasingly recognized as a key factor in the persistence of symptoms in affected individuals. Specifically, the hypercoagulable state in PCS may contribute to ongoing symptoms by promoting microvascular clots and organ damage. These microclots can impair circulation in small blood vessels, especially in the lungs, brain, and heart, potentially leading to symptoms such as shortness of breath, brain fog, fatigue, and muscle pain. Additionally, chronic inflammation, often associated with hypercoagulability, can result in symptoms like joint pain, muscle aches, and overall fatigue, all of which are common complaints in PCS. Our findings on thrombin generation further underscore this prothrombotic state in PCS. Our study showed higher thrombin generation in PCS patients compared to healthy controls, but this increase was not as pronounced as in patients with acute COVID-19. Additional TGT tests also revealed a significantly reduced inhibition of thrombin production among PCS patients in comparison to healthy individuals. This finding may be associated with TM resistance, where the normal anticoagulant activity of TM is impaired, resulting in increased thrombin generation despite the presence of TM. The underlying mechanisms of TM resistance in TGT are multiple. One potential cause is inflammation and cytokine modulation, where conditions such as chronic inflammation may indirectly affect TM function by altering coagulation dynamics [17]. Another possibility is an imbalance in coagulation factors, where elevated levels of procoagulant factors (e.g., factor VIII) or decreased levels of natural anticoagulants (e.g., protein C, protein S, or antithrombin) may overwhelm TM’s effect. Increased levels of factor VIIIa saturate the ability of activated protein C (APC) to bind and cleave it, and so inducing APC resistance. Further studies are necessary to clarify the mechanisms contributing to TM resistance in PCS patients.

Such thrombotic imbalances foster microthrombosis, a state akin to secondary thrombotic microangiopathy [18,19]. Investigations into coagulation profiles have highlighted a heightened VWF(Ag)/ADAMTS13 ratio, especially in severe COVID-19 cases [20]. We extended analysis of the VWF(Ag)/ADAMTS13 ratio to patients with PCS. Unlike during acute infection, where the VWF(Ag)/ADAMTS13 ratio has been shown to reach median levels of around 6.07 [6], our PCS cohort did not present an increased ratio (median of 0.91). These findings therefore suggest that the high VWF(Ag)/ADAMTS13 ratio observed in patients with acute COVID-19 infection may settle over time, despite ongoing symptoms of PCS, with the ratio returning to normal in most patients. However, a limitation is that longitudinal measurements repeated over time are lacking in this study. Although we could not demonstrate a higher level of vWF in all PCS patients, 12.4% did have elevated VWF levels, ranging from 150 to 226 IU/dL. Among those patients, all besides one exhibited a VWF(Ag)/ADAMTS13 ratio ≥ 1.5 (11.3%), a threshold potentially indicating an elevated thrombotic risk. Additionally, the group with a ratio ≥ 1.5 was significantly older, and all these patients presented with hypoperfusion in the frontal lobes or perfusion heterogeneity on cerebral perfusion scintigraphy. Additionally, it is noteworthy that every patient with a ratio ≥ 1.5 exhibits significant alterations in cerebral perfusion patterns with hypoperfusion in the frontal lobes or perfusion heterogeneity on cerebral perfusion scintigraphy. Further investigation is needed to determine whether there is a correlation between the ratio and the observed perfusion changes. These perfusion issues in brain areas, such as frontal lobe hypoperfusion, as described by Ajčević et al. [21], may link the observed hypercoagulable state to cognitive dysfunction reported in PCS. Furthermore, it would be important to investigate whether these patients also demonstrate more pronounced cognitive impairment. In addition to cognitive dysfunction, poor exercise tolerance is common in PCS patients. Prior studies, such as that of Prasannan et al. [22], linked a higher VWF(Ag)/ADAMTS13 ratio to diminished exercise capacity, where 55% of patients with impaired exercise tolerance showed a ratio ≥ 1.5.

In addition to comparing PCS patients with healthy subjects and acute COVID patients, it would have been valuable to include a cohort of convalescent COVID-19 patients who do not exhibit persistent symptoms (PCS -). The sample size for acute COVID-19 patients is limited, but all individuals on anticoagulants were excluded, so as to most closely reflect the acute phase of COVID-19.

## 5. Conclusions

In summary, our study revealed elevated in vitro thrombin generation and reduced thrombin inhibition in PCS patients, which could suggest an imbalance in coagulation factors or inflammation and cytokine modulation. Although most coagulation parameters appear to normalize over time, the persistence of abnormal VWF/ADAMTS13 ratios in a subset of patients points to a need for ongoing monitoring and potential antithrombotic intervention. Further studies are essential to elucidate the molecular pathways underpinning these findings and to evaluate the potential clinical significance of VWF/ADAMTS13 ratios in managing PCS. Such insights could enhance our understanding of PCS and support the development of targeted therapies to mitigate the vascular complications associated with this condition.

## Figures and Tables

**Figure 1 jcm-14-00789-f001:**
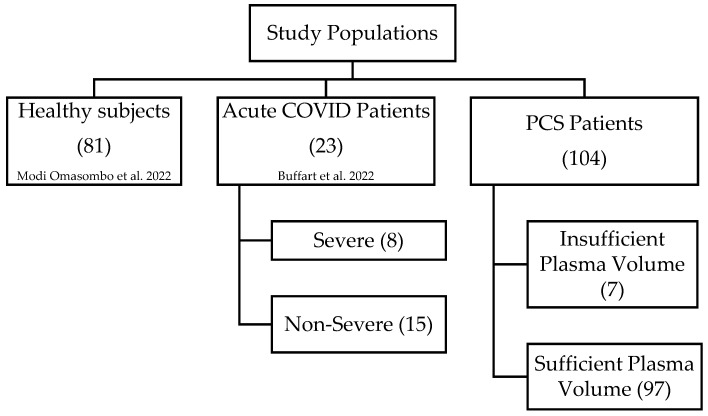
Hierarchical representation of the study populations [4,12].

**Figure 2 jcm-14-00789-f002:**
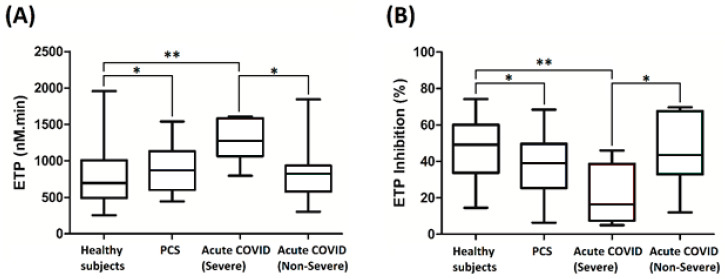
Comparisons among healthy subjects (*n* = 81), PCS patients (*n* = 97), acute COVID severe cases (*n* = 8), and acute COVID non-severe cases (*n* = 15) including the following: (**A**) endogenous thrombin potential with TM (ETP TM+); (**B**) endogenous thrombin potential inhibition. Data are presented as the median and the interquartile range. Comparisons between groups were assessed using Dunn’s post hoc test following a Kruskal–Wallis test (* *p* < 0.1, ** *p* < 0.01).

**Table 1 jcm-14-00789-t001:** Demographic parameters of acute and post-COVID syndrome (PCS) cohorts.

	Acute COVID		PCS
	Severe	Non-Severe	
Patients analyzed	8	15	97
Age	48–74	46–71	22–80
Median age	50	53	48
Male—*n* (%)	3 (37.5)	7 (46.7)	21 (21.6)

**Table 2 jcm-14-00789-t002:** Thrombin generation test results among historical cohorts as previously described and PCS. Results are expressed as medians and interquartile ranges. Comparisons between groups were assessed by Kruskal–Wallis test followed by Dunn’s post hoc test with PCS (* *p* < 0.1, ** *p* < 0.01, *** *p* < 0.001). Abbreviations: NS, not significant; TM, thrombomodulin; PCS, post-COVID syndrome.

	Healthy Subjects (*n* = 81)	Severe (*n* = 8)	Non-Severe (*n* = 15)	PCS (*n* = 97)	Kruskal–Wallis Test *p* Value	Dunn’s Post Hoc Test		
						PCS vs. Healthy Subjects	PCS vs. Acute COVID	
							Severe	Non- Severe
Without TM								
Lag time (min)	1.3	1.3	1.3	1.1	<0.001	***	NS	NS
	(1.1–1.5)	(1.1–1.5)	(0.99–1.6)	(1.0–1.3)				
Peak height (nM)	88	133	124	95	<0.001	NS	*	NS
	(66–113)	(129–170)	(100–137)	(79–127)				
Time to peak (min)	1.3	1.1	1.1	1.2	0.067	NS	NS	NS
	(1.1–1.5)	(1.0–1.2)	(0.87–1.5)	(1.1–1.4)				
ETP (nM.min)	102	109	105	105	0.191	NS	NS	NS
	(82–114)	(102–128)	(83–122)	(91–121)				
Velocity index (nM/min)	78	173	122	82	<0.001	NS	**	NS
	(51–111)	(124–229)	(88–169)	(61–121)				
With TM								
Lag time (min)	3	2.7	2.8	2.7	0.005	**	NS	NS
	(2.7–3.5)	(2.3–3.2)	(2.3–3.5)	(2.4–3.0)				
Peak height (nM)	143	307	173	166	0.002	NS	*	NS
	(99–210)	(230–341)	(139–225)	(120–232)				
Time to peak (min)	5.3	4.7	4.8	5.1	0.090	NS	NS	NS
	(4.7–6.0)	(4.1–5.2)	(3.9–6.1)	(4.6–5.5)				
ETP (nM.min)	695	1274	831	866	<0.001	*	NS	NS
	(490–1006)	(1064–1583)	(578–932)	(604–1131)				
ETP inhibition (%)	49	16	43	39	<0.001	**	NS	NS
	(34–60)	(7.6–39)	(33–68)	(25–50)				
Velocity index (nM/min)	85	222	126	92	<0.001	NS	**	NS
	(54–136)	(143–246)	(84–185)	(60–138)				

**Table 3 jcm-14-00789-t003:** Normal range and median values obtained for VWF(Ag), ADAMTS-13 activity, and VWF(Ag)/ADAMTS13 ratio in PCS patients.

	Normal Range	Median (IQR)
VWF(Ag)	50–150 IU/dL	98 (77–127)
ADAMTS-13 activity	50–150 IU/dL	105 (97–112)
VWF(Ag)/ADAMTS13 ratio	~1	0.91 (0.73–1.3)

**Table 4 jcm-14-00789-t004:** VWF(Ag)/ADAMTS13 ratios and median TGT results using Mann–Whitney test. Abbreviations: NS, not significant; ETP, endogenous thrombin potential.

	VWF(Ag)/ADAMTS13 Ratio		Mann–Whitney Test
	<1.5 *n* = 86	≥1.5 *n* = 11	
Age (years)	47	54	*p* < 0.005
BMI (kg/m^2^)	26	29.7	NS
Without thrombomodulin			
Peak height (nM)	200.9	239.5	NS
ETP (nM/min)	1388	1471	NS
With thrombomodulin			
Peak height (nM)	163.1	172	NS
ETP (nM/min)	862.8	1051	NS
ETP inhibition (%)	39.39	36.91	NS

## Data Availability

The data presented in this study are available on request from the corresponding author. The data are not publicly available due to patients’ privacy.

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
