# Peer review of "Exploring Hypercoagulability in Post-COVID Syndrome (PCS): An Attempt at Unraveling the Endothelial Dysfunction"

_jcm, 2025, doi:10.3390/jcm14030789_

Round 1
Reviewer 1 Report
Comments and Suggestions for Authors
Muys et al compared thrombin generation (with and without soluble thrombomodulin) and the von Willebrand factor (VWF)/ADAMTS13 axis among 97 PCS patients, 81 healthy controls and 23 acute COVID patients. PCS patients were found to have elevated thrombin production in the presence of TM, and a subset of patients exhibited elevated VWF/ADAMTS13 ratios that were associated with cognitive impairments and brain hypoperfusion. This study presents results that is of interest and with proper sample size, but the result description and discussion should be significantly modified.
Comments:
The source of the controls and acute COVID patients should be briefly disclosed in the method section, with proper reference.
The results presented in table 2 should be checked: I believe the unit of TG parameters from ST-Genesia usually is presented as % of a normal control plasma, instead of absolute nM thrombin.
Check table 4, both TGs were labeled as “without thrombomodulin”; how many patients were divided into each group?
Lines 172-173 claimed that patients with an abnormal ratio showed hypoperfusion in the frontal lobes or perfusion heterogeneity, but no data were presented anywhere, nor were the definitions of these claims.
The “soluble TM-resistant in vitro TG” phenotype in PCS could be multi-factorial: it could be due to deficiency of protein C and/or protein S, or due to elevated FVIII etc. The authors did not adequately investigate or discuss the possible underlying mechanisms. The statement of “impaired thrombomodulin function” in the abstract is false.
The claim in the abstract of an association between abnormal vwf/adamts13 ratio and cognitive impairments, and brain hypoperfusion lack support of any experiment data, it seems like a pure speculation.
Reviewer 2 Report
Comments and Suggestions for Authors
Authors evaluated coagulation disorders in patients diagnosed with PCS through a detailed analysis of thrombin generation and endothelial health markers. By using thrombin generation test with TM, alongside assessments of VWF antigen and ADAMTS-13 activity, we seek to investigate the extent of coagulation abnormalities and endothelial dysfunction in PCS patients.
Although this manuscript is potentially interesting, several issues arise.
1. Authors should compare PCS (+) with PCS (-) after COVID-19.
2. Patients number of acute COVID-19 is not large.
3. What is the definition for severe COVID-19? Is that severe illness or critical illness?
4. Relationship between hypercoagulability and symptom from PCS should be discussed.
5. Has the thrombotic risk been reported to be high in PCS?
6. There was no result which suggested conclusion.
7. There was no explanation for ETP.
8. Information such as weight, BMI and underlying disease may be helpfu.
Round 2
Reviewer 1 Report
Comments and Suggestions for Authors
i'd like to thank the authors for addressing my previous questions, i have no more comments.
Reviewer 2 Report
Comments and Suggestions for Authors
Revised paper has been partially improved.
Study size was still small.
Definition of severity was not clear.
